# Hollow-Channel Paper Analytical Devices Supported Biofuel Cell-Based Self-Powered Molecularly Imprinted Polymer Sensor for Pesticide Detection

**DOI:** 10.3390/bios12110974

**Published:** 2022-11-05

**Authors:** Yanhu Wang, Huihui Shi, Jiantao Sun, Jianjian Xu, Mengchun Yang, Jinghua Yu

**Affiliations:** 1Shandong Analysis and Test Center, Qilu University of Technology (Shandong Academy of Sciences), Jinan 250014, China; 2School of Chemistry and Chemical Engineering, University of Jinan, Jinan 250022, China; 3Shandong Institute for Product Quality Inspection, Jinan 250102, China; 4Department of Food and Drug, Weihai Ocean Vocational College, Weihai 264300, China

**Keywords:** self-powered, biofuel cell, hollow-channel, paper-based analytical device, molecularly imprinted polymers

## Abstract

Herein, a paper-based glucose/air biofuel cell (BFC) was constructed and implemented for self-powered pesticide detection. Our developed paper-based chip relies on a hollow-channel to transport fluids rather than capillarity, which reduces analysis times as well as physical absorption. The gold nanoparticles (Au NPs) and carbon nanotubes (CNTs) were adapted to modify the paper fibers to fabricate the flexible conductive paper anode/cathode electrode (Au–PAE/CNT–PCE). Molecularly imprinted polymers (MIPs) using 2,4-dichlorophenoxyacetic acid (2,4-D) as a template were synthesized on Au–PAE for signal control. In the cathode, bilirubin oxidase (BOD) was used for the oxygen reduction reaction. Based on a competitive reaction between 2,4-D and glucose-oxidase-labeled 2,4-D (GOx-2,4-D), the amount of GOx immobilized on the bioanode can be simply tailored, thus a signal-off self-powered sensing platform was achieved for 2,4-D determination. Meanwhile, the coupling of the paper supercapacitor (PS) with the paper-based chip provides a simple route for signal amplification. Combined with a portable digital multi-meter detector, the amplified signal can be sensitively readout. Through rational design of the paper analytical device, the combination of BFC and PS provides a new prototype for constructing a low-cost, simple, portable, and sensitive self-powered biosensor lab-on-paper, which could be easily expanded in the field of clinical analysis and drug delivery.

## 1. Introduction

Paper, as the most accessible medium for messages and knowledge transport throughout human history, has attracted considerable attention in the field of point-of-care testing (POCT) [1]. The distinct advantages of being abundant, low-cost, and easily disposable, enable paper to be a promising solution to develop biosensors with different detection techniques in resource-limited regions [2,3]. In recent years, microfluidics paper-based analytical devices (μ-PADs) for POCT have become increasingly popular and have attracted more and more interest, and various analytical methods have been established on μ-PADs for quantitative analysis [4]: such as colorimetric [5], fluorescence [6], electrochemical [7], chemiluminescent [8], electrochemiluminescent [9], and photoelectrochemical [10,11]. Although, the developed μ-PADs still featured the advantages of paper, most of them relied on the capillary action of porous paper channels to transport fluids that was time-consuming and allowed physical absorption: therefore, restricting its popularization [12,13]. Recently, the emergence of hollow–channels through removing the cellulose matrix opens up new insights to improve mass transfer efficiency and alter physical absorption, which is beneficial to their popularization in the field of POCT [14,15,16].

Pesticides have been widely used in the agricultural industry to improve crop yields [17]. However, the abuse of pesticides has also brought great detriment to the ecosystem and human health due to their toxicity [18]. Especially, 2,4-dichlorophenoxyacetic acid (2,4-D) has been widely utilized to eliminate broadleaf weeds in the field crops. Their structural stability makes them resistant to degradation and more likely to accumulate, which may lead to cancer in humans, and endocrine-disrupting issues [19]. Therefore, it is desirable to develop a simple and sensitive method to realize 2,4-D monitoring. Until now, the common methods for 2,4-D assays have been gas chromatography–mass spectrometry and liquid chromatography–mass spectrometry [20,21]. However, the time-consuming procedures, sophisticated instruments, and limited selectivity have restricted their extensive application. As synthesized artificial receptor molecules, molecularly imprinted polymers (MIPs) possess instinctive mechanical/chemical stability, a low-cost that has distinguished MIPs from other biometric recognition molecules, and they have attracted considerable attention among the various biosensors [22,23,24,25,26,27,28,29].

The introduction of MIPs into μ-PADs provides a new analytical approach for pesticide detection [30]. However, most methods fabricated on μ-PADs such as electrochemical and electrochemiluminescence, usually need to be excited by external energy sources causing them to depart from on-site monitoring [31,32]. Considering the stated problems, great efforts have been devoted to exploit portable power sources that simplify complicated operation [33]. Great achievements have been made, but there are still great improvements possible. Biofuel cells (BFCs), harvest electric from chemical energy, representing a new type of green energy conversion device based on biocatalysts [34,35]. Furthermore, the mild operation condition, miniaturization and easy integration of energy conversion with analytes offer a promising approach for self-powered on-site analysis [36]. Various self-powered biosensors based on BFCs have been fabricated and successfully implemented in the field of diagnosis and environmental monitoring [37,38,39].

Despite great progress having been achieved, most of the BFC-based self-powered sensors rely on sophisticated electrochemical workstations for signal collection that departs from the development of portable, integration and low-cost systems Fortunately, coupling BFCs with capacitors provides an effective way to enable high power charge/discharge cycles as well as for the output signal amplification [40,41]. Based on this, we demonstrated a self-powered biosensor that integrated glucose/O_2_ BFC and an all-solid-state paper supercapacitor (PS) within a signal piece of paper based on origami for 2,4-D detection (Figure 1). The fabricated BFC is composed of a glucose oxidase (GOx) bioanode and a bilirubin oxidase (BOD) biocathode. MIPs grafted on the bioanode of BFC through electropolymerization of polypyrrole (PPy) acted as biological receptors for specific recognition of 2,4-D. The competition between free 2,4-D and GOx labeled 2,4-D (Gox–2,4-D), further influence the electric output of BFC. The generated current was temporarily collected and stored within the PS realizing an amplified current output detected by a digital multimeter (DMM). On the other hand, hollow channels were designed on the paper device to eliminate the barriers regarding low fluid flow rates and significant nonspecific adsorption. Overall, all these features accounted for excellent analytical performance toward 2,4-D. Our developed method paves a new way for developing integrated, portable, and versatile paper-based analytical devices, as well as provides a new choice for diagnosis and environmental monitoring in resource-limited regions.

## 2. Experimental Section

### 2.1. Fabrication of the Origami Biofuel Cell-Based Lab-on-Paper Device (BFCPD)

In this work, the sketch of the origami biofuel cell-based lab-on-paper device (BFCPD) was designed on Adobe illustrator CS6. Different functional zones were differentiated by color. The configuration of the constructed BFCPD comprised an anodic tab (blue), a hollow-channel tab, a cathodic tab (dark blue), a hemichannel tab (green), and two paper supercapacitor tabs (Appendix A). The square hydrophilic zone (2.5 × 2.5 mm) on the hollow-channel tab and cathodic tab, named as the “via hole”, was modified by the Au NPs’ decorated multiwalled carbon nanotubes (Au–MWCNTs) that enable the “via hole” electrical conduction from bottom to top. Each tab was folded based on the unprinted line. The unprinted rectangle region was used for the fabrication of PS (Appendix A). After wax printing, the obtained paper sheet was baked at 130 °C until the wax melted and penetrated through the paper to form the hydrophobic and insulating patterns (Appendix A). Subsequently, the paper sheet was ready for carbon electrodes and the silver conductive wires screen-printing (Appendix A). A light yellow hemichannel on the hemichannel tab was obtained through varying the values of C, M, Y, and K. The hydrophilic paper fibers on the cathodic tab and the hollow–channel tab and reservoirs were removed. Finally, the PS was fabricated based on reported work [3]. Specifically, the hydrophilic zone of the anodic/cathodic tab (part a in Appendix A) was functionalized with Au nanoparticles and ionic liquid CNTs endowing the paper electrode with superior conductivity to fabricate Au–PAE (part b in Appendix A) and CNT-PCE (part b in Appendix A). The PS was fabricated based on the reported work [42,43].

### 2.2. Fabrication of Bioanode

The MIPs grafted on Au-PAE were fabricated through electropolymerization of polypyrrole (PPy) MIPs with the presence of the template molecule (2,4-D) (part c in Appendix A). An electrolyte solution containing 0.1 M KCl, 0.5 mM pyrrole, and 10 mM 2,4-D in 0.05 M phosphate buffer solution (PBS, pH 7.0) was first prepared and deoxidized in bubbling nitrogen gas for 10 min. Subsequently, the electropolymerization was performed at a constant voltage of 0.8 V (vs. Ag/AgCl). After a 10-min reaction, the template molecules were removed and rinsed through a methanol and acetic acid mixture (70% methanol, *v*/*v*). Afterwards, the obtained MIP–Au–PAE was rinsed with ultrapure water and dried in an air stream. Meanwhile, the non-imprinted polymers (NIPs)-grafted Au–PAE was prepared with the same procedure except for the introduction of template molecules into the non-imprinted polymers (NIPs)-grafted Au–PAE.

### 2.3. Fabrication of Biocathode

This process involved 10 µL of BOD (10 μg·mL^−1^, dissolved in 0.1 M pH 7.4 PBS) being cast onto the CNT-PCE and maintained for 2 h at room temperature (part c in Appendix A). Subsequently, physically absorbed molecules were removed through washing with PBS (0.01 M, pH 7.4). The resulting biocathodes (BOD–CNT–PCE) were stored at 4 °C prior to use.

### 2.4. Assay Procedure

Prior to assay, the designed BFCPD was first folded as indicated in Appendix A, and then clamped by a device-holder guaranteeing that the BFCPD was stacked closely. After that, 50 μL of a 10 mM 2,4-D solution was introduced into the BFCPD through the inlet, and the fluid transported along the designed hollow channel finally reached the bioanode. After a 25-min incubation, the recognition cavities on the MIPs were fully blocked. Excess solution was removed from the inlet. Subsequently, 50 μL of 10 mM GOx-2,4-D solution was added and incubated for another 25 min. During this process, the GOx-2,4-D occupied the binding sites leading to replacement of 2,4-D (part d in Appendix A). Part of the 2,4-D could not be replaced due to the stereoscopic hindrance effect from GOx. To assess the sample detection, a 30 μL aqueous solution with different concentrations of 2,4-D was added and incubated for 25 min. A competitive reaction occurred, the GOx2,4-D was replaced by the free 2,4-D (part e in Appendix A). After washing with PBS, 50 µL of 30 mM glucose solution was added and migrated to the bioanode and biocathode to initiate the reaction. The generated current was collected and stored within the PS temporarily. After 60 s charging, a high current intensity could be detected by the DMM. Based on the amplified current intensity, the 2,4-D could be quantified.

## 3. Results and Discussion

### 3.1. Morphology Characterization

Figure 1A,B shows the scanning electron microscope (SEM) images of the bare paper electrode which possesses porous architecture and a rough surface. Compared with the bare paper, continuous and dense AuNPs assembled on the fibers can be seen (Figure 1C), and the paper still maintains the original porous microstructure (Figure 1D). The SEM image in Figure 1E,F presents well-dispersed CNTs on the surface of the paper fibers, also demonstrating the successful fabrication of CNT–PCE. The decoration of AuNPs and CNT would improve the conductivity that accounted for the superior performance of BFC. To investigate the morphology of the fabricated MIPs–Au–PAE, the SEM images were derived and the results are shown in Figure 1G,H. Compared with Au–PAE (shown in Figure 1C,D), the surface of fibers is rougher after the decoration of PPy MIPs, and the diameter increases with the electro-polymerized of PPy MIPs, which also illustrates a successful decoration of PPy MIPs on the surface of Au-paper fibers.

### 3.2. Electrochemical Characterization of the Bioanode and Biocathode

The electrochemical behavior of the stepwise fabrication process of MIPs–Au–PAE and BOD–CNT–PCE was investigated through cyclic voltammetry (CV) toward a 10.0 mM [Fe(CN)_6_]^3−/4−^ solution containing 0.5 M KCl. As shown in Figure 2A, compared with the bare PAE (curve a), a higher current response could be observed after the coating of a Au layer on paper fibers (curve b), which was attributed to the factor that the Au layer could greatly increase the conductivity. After the MIPs were grafted on the Au–PAE (curve c), a remarkable decrease in the peak current was detected due to the reduced conductivity of MIPs layers that acted as a definite kinetic barrier for the charge transfer. The current increased accordingly after the removal of the template (curve d), which suggested that the [Fe(CN)_6_]^3−/4−^ molecules were more easily available for the electrode. After template rebinding, the current decreased again (curve e). This may be caused by the molecules rebinding having blocked the diffusion of [Fe(CN)_6_]^3−/4−^ to the surface of the electrode. These results verified the successful construction of the bioanode. The results obtained from the electrochemical impedance spectroscopy (EIS) (Figure 2B) also supported the above results.

Meanwhile, CV (Figure 2C) and EIS (Figure 2D) were also selected to explain the fabrication procedure of the biocathode. Specifically, the CV response of different electrodes in the sequence of PCE < BOD–CNT–PCE < CNT–PCE, implied the successful construction of the BOD–CNT–PCE. Moreover, the variation of the semicircle in EIS (Figure 2D) also reflected the successful fabrication of the biocathode.

Electrochemical measurements were carried out to evaluate the performance of fabricated bioanode and biocathode. Figure 3A shows that the bioelectrocatalytic current resulted from the enzymatic oxidation of glucose within a voltage range from −0.6 to 0.4 V versus Ag/AgCl. A low glucose oxidation potential of around 0.19 V (vs. Ag/AgCl) was observed, which is consistent with the fact that the GOx comprises two subunits that each include the flavin adenine dinucleotide (FAD) cofactor [44]. Figure 3B displays the polarization curves of the prepared bioanode (incubation with 100 ng·mL^−1^ 2,4-D) in the presence of 30 mM glucose. With the potential scanning from −0.6 V to 0.8 V, the anodic currents increased correspondingly. It should be observed that an obvious catalytic electrooxidation potential at −0.2 V (vs. Ag/AgCl) for glucose was observed, and reached a plateau current density of about 10 mA·cm^−2^ near −0.1 V (vs. Ag/AgCl) (curve b) compared to the bare electrolyte (curve a). This result reveals the high electrocatalytic activity of GOx toward glucose oxidation at the MIP-Au-PAE.

The performance of the fabricated biocathode was also studied in 0.1 M PBS (pH 7.4, containing 0.1 M NaCl) saturated by O_2_ or N_2_ (Figure 3C). From this comparison, the onset potential around 0.5 V for O_2_ reduction is similar to the thermodynamic equilibrium potential of E^ϴ^ O_2_/H_2_O (0.61 V at pH 7.0) with the absence of O_2_ (curve b) [34]. However, there was no peak observed for BOD–CNT–PCE in N_2_-saturated solution (curve a). Figure 3D shows the polarization curves of BOD–CNT–PCE in a N_2_ (curve a) -saturated solution, and an O_2_ (curve b) -saturated solution. It could be clearly seen that the BOD–CNT–PCE displays a higher current density in an O_2_-saturated surrounding than that in N_2_-saturated condition. The above results clearly demonstrate the superior performance of the BOD–CNT–PCE toward oxygen reduction reaction.

### 3.3. General Working Principle of the Assembled BFC

The general working principle of the assembled BFC is illustrated in Figure 1. Briefly, the GOx fix on the MIP–Au–PAE could efficiently catalyze the oxidation of glucose selectively to generate gluconolactone. At the same time, the FAD within GOx was reduced to GOx–FADH_2_ [45]. The GOx–FADH_2_ automatically released two H^+^ and electrons, and regenerated to GOx–FAD smoothly, leading to the efficient electron transfer from GOx to Au–PAE and the electron flow from bioanode to biocathode through the external circuit. The O_2_ molecules are reduced to H_2_O under a four proton-assisted electron transfer process at the biocathode with the participation of H^+^, eventually realizing the energy conversion from chemical energy to electric energy. The whole process can be illustrated in the following reaction equations:$$\mathrm{GOx}-\mathrm{FAD}+\mathrm{glucose}\to \mathrm{GOx}-{\mathrm{FADH}}_{2}+\mathrm{gluconolactone}$$
$$\mathrm{GOx}-{\mathrm{FADH}}_{2}\to \mathrm{GOx}-\mathrm{FAD}+2H^{+}+2e^{-}$$
$$O_{2}+4H^{+}+4e^{-}\overset{\mathrm{BOD}}{\to }2H_{2}O$$

### 3.4. Optimization of Experimental Conditions

In order to obtain optimal analytical performance and efficiency, the incubation time was optimized. First, the absorption time between the eluted MIPs–Au–PAE and 2,4-D was investigated. As shown in Appendix A, the residual 2,4-D in solution decreased quickly with extending the absorption time, and reached a plateau after 25 min, indicating the equilibrium rebinding between the MIPs and the template. Therefore, 25 min was selected as the optimal absorption time in this work. Similarly based on the results in Appendix A, 25 min was also used as the optimal GOx–2,4-D labeling time and the competitive reaction time between GOx–2,4-D and 2,4-D.

### 3.5. Analytical Performance

Under optimal conditions, the quantitative analysis and dynamic range of the developed BFCPD toward 2,4-D were evaluated by varying 2,4-D concentration in a standard solution. As expected, good linear range relationships between the detected current without (Figure 4A) or with (Figure 4B) the PS amplification and the logarithmic concentration of 2,4-D was observed in the range from 1.0 pM-50.0 μM. The fitted linear equations were *I* = 74.20–9.68l g [2,4-D/pM] (R = 0.9946) and *I_PS_* = 949.22–122.18l g [2,4-D/pM] (R^2^ = 0.985). The limit of detection (LOD) was estimated to be 0.53 pM based on the signal-to-noise (S/N) ratio of 3. It is interesting to see that the LOD obtained with or without PS are the same, which might be mostly attributed to the concomitant amplification of the background. Moreover, it should be observed that the instantaneous amplified current discharged from PS is about 13-fold higher than that without PS charging. The amplified current could be detected through the DMM, causing the complicated electrochemical workstation to be abandoned.

Subsequently, the power density of the assembled glucose/O_2_ BFC after incubation with 1.0 μM 2,4-D in 0.10 M PBS (pH 7.0) containing 30 mM glucose was measured. At the specified conditions, the fabricated BFC exhibited an open-circuit voltage (Voc) of 0.75 V, and the maximum power density (P_max_) reached 152 mW·cm^−2^ at 0.5 V.

### 3.6. Specificity, Reproducibility, and Stability Investigation

Selectivity is a crucial standard to evaluate the performance of fabricated biosensors. Thus, 2,4,5-trichlorophenoxyacetic acid (2,4,5-T), 4-chlorophenoxyacetic acid (CPA), 2,4-dichlorophenol (2,4-DCP), 4-(2,4-dichlorophenoxy) butyric acid (2,4-DBA), and 4-chloro-o-tolyloxyacetic acid (MCPA) were selected as interferents to access the specificity of our developed BFCPD. As shown in Figure 4D, those interferents could barely influence the signal output and were similar to the blank. Only in the presence of 2,4-D, could a dramatically decreased current be detected. Those results demonstrated that the constructed biosensor possessed satisfied selectivity. Additionally, the reproducibility was investigated through intra- and inter-assays under the same conditions. An identical signal response with the relative standard deviation (RSD) of 3.4% and 5.2% indicated a gratifying reproducibility. Furthermore, the stability of the developed BFCPD was evaluated through measuring the current intensity at intervals of five days. No significant changes of the current output were observed when stored in a refrigerator at 4 °C for four weeks, revealing the good stability of the developed BFCPD.

### 3.7. Practical Application

Finally, the recovery test was conducted to evaluate the feasibility of the fabricated biosensor toward a practical application. The standard addition method was employed by adding different concentrations of 2,4-D into tap water and lake water, respectively. As shown in Table 1, the recovery values of the constructed biosensor were obtained in the range of 98.3-104% for tap water, and 97.5-103.5% for lake water. Together with the RSD less than 3.2% and 4.3%, this validated the reliability and practicability of our developed BFCPD for 2,4-D determination in real samples.

## 4. Conclusions

In this work, a sensitive paper-based self-powered sensing protocol based on a glucose/O_2_ BFC device integrated with a molecularly imprinted technique for 2,4-D detection was fabricated for the first time. A PS constructed on the fabricated BFCPD and a DMM were used as the current amplifier and the terminal current detector, respectively. Meanwhile, hollow channels were introduced into the BFCPD to transport fluids which could make them suitable for point-of-care testing. The construction of the conductivity of PPy MIPs layers offers a promising approach for 2,4-D-specific recognition. This is based on a competitive reaction between 2,4-D and GOx–2.4-D, and the influence of the catalytic oxidation of glucose at the bioanode to realize quantification. Meanwhile, the introduction of PS could collect and store the generated electrons leading to an amplified current response that can be directly read by the portable DMM. This work provides a novel and efficient approach to develop a multifunctional paper device to implement self-powered sensing, that eliminates the energy source. Such findings also provide an alternative choice for the design and development of POCT devices.

## Data Availability

Not applicable.

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
