# Peer review of "Hollow-Channel Paper Analytical Devices Supported Biofuel Cell-Based Self-Powered Molecularly Imprinted Polymer Sensor for Pesticide Detection"

_biosensors, 2022, doi:10.3390/bios12110974_

Round 1

Reviewer 1 Report

The authors reported a paper-based analytical device integrated with biofuel cell to realize simple, sensitive, accurate and self-powered pesticide assay. Moreover, the combination with a paper supercapacitor, an amplified detection signal was obtained. Overall, this work has very good novelty, the results were well organized, and the manuscript was written in a straightforward manner. This study is likely to appeal to interested readers of Biosensors. Minor revisions as specified below are needed before acceptance.

(1) The strategy of the sensing detection should be explained more clearly for readers.

(2) Please explain how to calculate the detection limit in your sensing system in detail.

(3) Why the sensitivity with and without paper supercapacitor both have the same limit of detection. How can the sensitivity be improved without an improvement in the LOD?

(4) The scan direction should be presented in in Fig. 2.

(5) There are some grammatical or spelling errors in the manuscript, which should be revised carefully.

(6) Would different operators generate variation on the detection of the same sample with same assay?

(7) Some recent references related to molecular imprinting polymer based sensors are recommended to be cited, such as Materials Today Chemistry 26 (2022) 101043; Journal of Hazardous Materials 436 (2022) 129107; Analytica Chimica Acta, 2022, 1200, 339599.

Author Response

To the comments from referee 1:

The authors reported a paper-based analytical device integrated with biofuel cell to realize simple, sensitive, accurate and self-powered pesticide assay. Moreover, the combination with a paper supercapacitor, an amplified detection signal was obtained. Overall, this work has very good novelty, the results were well organized, and the manuscript was written in a straightforward manner. This study is likely to appeal to interested readers of Biosensors. Minor revisions as specified below are needed before acceptance.

Reply:

Thank you very much for your kindly help and review. These comments are very valuable for us to improve the quality of our manuscript. Now the manuscript was revised carefully according to referees' comments. And details presented below.

(1) The strategy of the sensing detection should be explained more clearly for readers.

Reply:

Thank you very much for your kindly help and review.

Herein, a hollow-channel paper-supported glucose/O2 biofuel cell (BFC)-based self-powered sensing platform for pesticides analysis was developed. The BFC device utilized gold nanoparticle and carbon nanotubes (CNTs)-modified paper fibers as the anodic and cathodic substrate to wire glucose oxidase (GOx) and bilirubin oxidase (BOD) for the fabrication of bioanodes and biocathodes. To implement an assay protocol, molecularly imprinted polymers (MIPs) grafted on the bioanode was implemented for the selective recognition target molecules (2,4-D). Based on a competition between free 2,4-D and GOx labeled 2,4-D, the amount of GOx immobilized on the bioanode can be facilely tailored that further influence the performance of the fabricated BFC (current output). Combined with a paper supercapacitor (PS), the generated current could be temporary stored within the PS to further amplify the electrochemical signal. Based on this design, the present biosensor shows high performance, especially for detection limit and sensitivity.

(2) Please explain how to calculate the detection limit in your sensing system in detail.

Reply:

Thank you very much for your kindly help and review.

The limits of detection were obtained at a signal-to-noise ratio of 3σ (where σ is the standard deviation of a blank solution, n = 11) [1-3].

Reference

  1. Zhongju Song, Ruo Yuan, Yaqin Chai, Wen Jiang, Huilan Su, Xin Che and Xiaoqi Ran. Biosens. Bioelectron. 2011, 26, 2776-2780.
  2. Dianping Tang, Juan Tang, Biling Su and Guonan Chen. J. Agric. Food Chem. 2010, 58, 10824-10830.
  3. Zhifeng Fu, Zhanjun Yang, Jinhai Tang, Hong Liu, Feng Yan and Huangxian Ju. Anal. Chem. 2007, 79, 7376-7382.

(3) Why the sensitivity with and without paper supercapacitor both have the same limit of detection. How can the sensitivity be improved without an improvement in the LOD?

Reply:

Thank you very much for your kindly help and review.

The limit of detection was defined as three times the standard deviation of the blank/background [Anal. Chem. 2011, 83, 8035] [Anal. Chem. 2012, 84, 2837]. Therefore, the current measurements from the BFC and from paper supercapacitor have the same limit of detection due to the concomitant amplification of the background photocurrent from the BFC. Thus, the enhancement of the sensitivity in this work was confirmed through the comparison of the slope of the linear dynamic range. As indicated in Figure 4, the sensitivity of this paper based analytical device with paper supercapacitor  was higher (~13-fold) than that without the amplification through paper supercapacitor.

(4) The scan direction should be presented in in Fig. 2.

Reply:

Thank you very much for your kindly help and review.

According to your requirement, we have given the scan direction in the linear sweep voltammograms in Figure 2 of the revised manuscript. As for the bioanode, the scan direction from low potential to high potential, whereas for the biocathode, the scan direction is opposite.

(5) There are some grammatical or spelling errors in the manuscript, which should be revised carefully.

Reply:

Thank you very much for your kindly help and review.

According to your requirement, the English has been carefully revised by a native English teacher in our school. Thank you again very much for you kindly help and review.

(6) Would different operators generate variation on the detection of the same sample with same assay?

Reply:

Thank you very much for your kindly help and review.

Yes, the variation on the detection of the same sample with the same assay may be generated mainly due to the manual operations of the timer and the switch by different users.

This variation between different users could be eliminated by fabricating a built-in and automatic timer and switch on this paper based analytical device.

(7) Some recent references related to molecular imprinting polymer based sensors are recommended to be cited, such as Materials Today Chemistry 26 (2022) 101043; Journal of Hazardous Materials 436 (2022) 129107; Analytica Chimica Acta, 2022, 1200, 339599.

Reply:

Thank you very much for your kindly help and review.

Corresponding references have been cited to strengthen this paper.

Reviewer 2 Report

In this paper, the authors reported a paper supported biofuel cell based self-powered sensor for detecting pesticide. Overall, the related characterizations are well conducted and this paper is of significance to a certain extent in engineering. However, before further consideration, some concerns should be well addressed, which are listed below for your reference.

1. The self-powered sensing mechanism should be presented more clearly. The current scheme confused readers. The different components should be indexed well.

2. The figures should be well re-organized. Some unimportant figures are suggested to transferred to Supporting Information.

3. The basic properties, including sensitivity, repeatability, stability, detection limit, and others, should be provided in the paper.

4. The data seems insufficient. Some deep investigations should be carried out to rich this paper. In addition, there is a lack of control experiments.

5. When mentioned the concept of self-powered, a few related articles might give you a helpful reference. You can cite them in the paper. Such as: DOI10.1021/acsami.1c12443.

Author Response

To the comments from referee 2:

In this paper, the authors reported a paper supported biofuel cell based self-powered sensor for detecting pesticide. Overall, the related characterizations are well conducted and this paper is of significance to a certain extent in engineering. However, before further consideration, some concerns should be well addressed, which are listed below for your reference.

Reply:

Thank you very much for your kindly help and review. These comments are very valuable for us to improve the quality of our manuscript. Now the manuscript was revised carefully according to referees' comments. And details presented below.

(1) The self-powered sensing mechanism should be presented more clearly. The current scheme confused readers. The different components should be indexed well.

Reply:

Thank you very much for your kindly help and review.

Herein, a hollow-channel paper-supported glucose/O2 biofuel cell (BFC)-based self-powered sensing platform for pesticides analysis was developed. The BFC device utilized gold nanoparticle and carbon nanotubes (CNTs)-modified paper fibers as the anodic and cathodic substrate to wire glucose oxidase (GOx) and bilirubin oxidase (BOD) for the fabrication of bioanodes and biocathodes. To implement an assay protocol, molecularly imprinted polymers (MIPs) grafted on the bioanode was implemented for the selective recognition target molecules (2,4-D). Based on a competition between free 2,4-D and GOx labeled 2,4-D, the amount of GOx immobilized on the bioanode can be facilely tailored that further influence the performance of the fabricated BFC (current output). Combined with a paper supercapacitor (PS), the generated current could be temporary stored within the PS to further amplify the electrochemical signal. Based on this design, the present biosensor shows high performance, especially for detection limit and sensitivity.

According to your comments, a schematic diagram of bioanode and biocathode fabrication process were presented as below. Different components have been well indexed. And the corresponding description have been added into the manuscript. I think together with the below scheme and scheme in the manuscript is helpful for readers to easier understand the mechanism.

Scheme S1 Schematic diagram of the fabrication of bioanode (A) and biocathode (B), different part in (A) represents PAE (a), Au-PAE (b), MIPs-Au-PAE (c), GOx-2,4-D occupied MIPs-Au-PAE (d), and detection of 2,4-D at MIPs-Au-PAE; different part in (B) represents PCE (a), CNT-PCE (b), and BOD-CNT-PCE (c).

(2) The figures should be well re-organized. Some unimportant figures are suggested to transferred to Supporting Information.

Reply:

Thank you very much for your kindly help and review.

Figure 3 in optimization of experimental conditions has been moved into supporting information.

(3) The basic properties, including sensitivity, repeatability, stability, detection limit, and others, should be provided in the paper.

Reply:

Thank you very much for your kindly help and review.

The corresponding sensitivity, repeatability, stability, and detection limit have been added into the section of 3.4 and 3.5, and presented as below:

Sensitivity: Well liner range relationships between the detected current without (Figure 4A) or with (Figure 4B) PS amplification and logarithmically concentration of 2,4-D were observed in the range from 1.0 pM-50.0 μM. And the fitted liner equations were I=74.20-9.68lg[2,4-D/pM] (R=0.9946) and IPS=949.22-122.18lg[2,4-D/pM] (R2=0.985).

Detection limit: The limit of detection (LOD) was estimated to be 0.53 pM based on the signal-to-noise (S/N) ratio of 3.

Repeatability: The reproducibility was investigated through intra- and inter-assay under same conditions. And identical signal response with the relative standard deviation (RSD) of 3.4 % and 5.2 % indicated a gratifying reproducibility.

Stability: The stability of the developed BFCPD were evaluated through measuring the current intensity at intervals of five days. No significant changes of the current output were observed stored in a refrigerator at 4 °C for four weeks, revealing the good stability of developed BFCPD.

(4) The data seems insufficient. Some deep investigations should be carried out to rich this paper. In addition, there is a lack of control experiments.

Reply:

Thank you very much for your kindly help and review.

According to your comments, the modification procedure of bioanode and biocathode were investigated and presented as below:

The electrochemical behavior of the stepwise fabrication process of MIPs-Au-PAE and BOD-CNT-PCE were investigated through cyclic voltammetry (CV) toward 10.0 mM [Fe(CN)6]3–/4 − solution containing 0.5 M KCl. As shown in Figure 2A, compared with the bare PAE (curve a), higher current response could be observed after the coating of Au layer on paper fibers (curve b), which was attributed the factor that Au layer could greatly increase the conductivity. After the MIPs grafted on the Au-PAE (curve c), a remarkable decrease in the peak current was detected due to the reduced conductivity of MIPs layers that acted as a definite kinetic barrier for the charge transfer. The current increased accordingly after the removal of template (curve d), which suggested that the [Fe(CN)6]3–/4− molecules are more easily available for the electrode. After template rebinding, the current decreased again (curve e). This may be caused by the molecules rebinding blocked the diffusion of [Fe(CN)6]3–/4− to the surface of electrode. These results verified that the the successful construction of bioanode. And the results obtained from the electrochemical impedance spectroscopy (EIS) (Figure 2B) also supported above results.

Meanwhile, CV (Figure 2C) and EIS (Figure 2D) were also selected to explain the fabrication procedure of biocathode. Specifically, the CV response of different electrode in the sequence of PCE < BOD-CNT-PCE < CNT-PCE, implying the successful construction of BOD-CNT-PCE. Moreover, the variation of the semicircle in EIS (Figure 2D) also reflect the successful fabrication of biocathode.

Figure 2 CV (A) and EIS (B) responses of (a) PAE, (b) Au-PAE, (c) MIPs-Au-PAE, (d) MIPs-Au-PAE after template removal, (e) MIPs-Au-PAE after template rebinding; CV (A) and EIS (B) responses of PCE (a), CNT-PCE (b), and BOD-CNT-PCE (c).

Blank contrast experiments have been performed in the investigation of selectivity. And no conspicuous changes were observed, illustrating the well analytical performance of our proposed method toward target detection.

(5) When mentioned the concept of self-powered, a few related articles might give you a helpful reference. You can cite them in the paper. Such as: DOI: 10.1021/acsami.1c12443.

Reply:

Thank you very much for your kindly help and review.

Corresponding references have been cited to strengthen this paper.

Reviewer 3 Report

There are some questions.

Did the authors use a different combination of nanomaterials?

Regarding medical use. What grounds do the authors have that allow them to assume that the results can be expanded in the field of clinical analysis and drug screening?

Given the simplicity of the described procedure for creating an analyzer, it can be noted that the analyzer can be considered as a new type of device for constructing a low-cost, simple, portable, and sensitive self-powered biosensor. 

There are errors in the work.

An error was made in the designation: The scan rate for polarization curves is 1 mV·s1 ., fig.2.

Author Response

To the comments from referee 3:

There are some questions.

(1) Did the authors use a different combination of nanomaterials?

Reply:

Thank you very much for your kindly help and review.

In this work, gold nanoparticles and carbon nanotubes were used to construct the substrate of bioanode and biocathode. And in our previous work, gold nanoparticles, carbon nanotubes, and platinum nanoparticles have been used to constructed the substrate.[1-4]

References

[1] Wang Y, Ge L, Wang P, et al. A three-dimensional origami-based immuno-biofuel cell for self-powered, low-cost, and sensitive point-of-care testing[J]. Chemical Communications, 2014, 50(16): 1947-1949.

[2] Wang Y, Zhang L, Cui K, et al. supported self-powered system based on a glucose/O2 biofuel cell for visual microRNA-21 sensing[J]. ACS applied materials & interfaces, 2019, 11(5): 5114-5122.

[3] Wang Y, Zhang L, Zhao P, et al. Visual distance readout to display the level of energy generation in paper-based biofuel cells: application to enzymatic sensing of glucose[J]. Microchimica Acta, 2019, 186(5): 1-9.

[4] Wang Y, Ge L, Ma C, et al. Self-powered and sensitive DNA detection in a three-dimensional origami‐based biofuel cell based on a porous Pt-paper cathode[J]. Chemistry-A European Journal, 2014, 20(39): 12453-12462.

(2) Regarding medical use. What grounds do the authors have that allow them to assume that the results can be expanded in the field of clinical analysis and drug screening?

Reply:

Thank you very much for your kindly help and review.

In this work, a kind of pesticide, 2,4-D was used as template for the synthesis of MIPs. During this process, the template could be substituted by others, such as protein, peptide, glycan, viruses, or even entire cells.[1] Therefore, the MIPs based sensors could be expanded to the fields of clinical analysis. As for the “drug screening”, we are sorry for the careless mistake, “drug screening” should be “drug delivery”. And this mistake has been corrected.  

References

[1] Romana Schirhagl. Bioapplications for Molecularly Imprinted Polymers[J]. Analytical Chemistry, 2014, 86, 250-261.

(3) Given the simplicity of the described procedure for creating an analyzer, it can be noted that the analyzer can be considered as a new type of device for constructing a low-cost, simple, portable, and sensitive self-powered biosensor.

Reply:

Thank you very much for your kindly help and review.

In this work, a hollow-channel paper-supported glucose/O2 biofuel cell (BFC)-based self-powered sensing platform for pesticides analysis was developed. The BFC device utilized gold nanoparticle and carbon nanotubes (CNTs)-modified paper fibers as the anodic and cathodic substrate to wire glucose oxidase (GOx) and bilirubin oxidase (BOD) for the fabrication of bioanodes and biocathodes. To implement an assay protocol, molecularly imprinted polymers (MIPs) grafted on the bioanode was implemented for the selective recognition target molecules (2,4-D). Based on a competition between free 2,4-D and GOx labeled 2,4-D, the amount of GOx immobilized on the bioanode can be facilely tailored that further influence the performance of the fabricated BFC (current output). Combined with a paper supercapacitor (PS), the generated current could be temporary stored within the PS to further amplify the electrochemical signal. Based on this design, this inexpensive, disposable and sensitive BFCPD furnish a feasible and robust tool for the pesticide detection.

(4) There are errors in the work.

An error was made in the designation: The scan rate for polarization curves is 1 mV·s1 ., fig.2.

Reply:

Thank you very much for your kindly help and review.

The mistake has been corrected.

Round 2

Reviewer 2 Report

this paper can be considered of acceptance.